# Thermogenic Activation of Adipose Tissue by Caffeine During Strenuous Exercising and Recovery: A Double-Blind Crossover Study

**DOI:** 10.3390/metabo15080517

**Published:** 2025-08-01

**Authors:** Dany Alexis Sobarzo Soto, Diego Ignácio Valenzuela Pérez, Mateus Rossow de Souza, Milena Leite Garcia Reis, Naiara Ribeiro Almeida, Bianca Miarka, Esteban Aedo-Muñoz, Armin Isael Alvarado Oyarzo, Manuel Sillero-Quintana, Andreia Cristiane Carrenho Queiroz, Ciro José Brito

**Affiliations:** 1Department of Physical Education, Federal University of Juiz de Fora, Governador Valadares 35010-900, Brazil; danysobarzo@santotomas.cl (D.A.S.S.); mateus.rossow@estudante.ufjf.br (M.R.d.S.); milena.leite@estudante.ufjf.br (M.L.G.R.); naiara.ribeiro@ufjf.br (N.R.A.); andreia.queiroz@ufjf.br (A.C.C.Q.); 2Escuela de Kinesiología, Facultad de Salud, Universidad Santo Tomás, Santiago 8320000, Chile; diegovalenzuela@santotomas.cl; 3Laboratory of Psychophysiology and Performance in Sports & Combats, School of Physical Education and Sport, Federal University of Rio de Janeiro, Rio de Janeiro 20080-003, Brazil; biancamiarka@eefd.ufrj.br; 4Escuela de Ciencias de la Actividad Física, Deporte y la Salud, Facultad de Ciencias Médicas, Universidad de Santiago de Chile, Santiago 8320000, Chile; esteban.aedo@usach.cl; 5Escuela de Kinesiología, Facultad de Salud, Universidad Santo Tomás, Puerto Montt 5480000, Chile; aalvarado13@santotomas.cl; 6Faculty of Physical Activity and Sport Sciences (INEF), Universidad Politécnica de Madrid, 28000 Madrid, Spain; manuel.sillero@upm.es

**Keywords:** caffeine, brown adipose tissue, infrared thermography, high-intensity interval exercise, metabolism

## Abstract

**Background/Objectives:** To investigate acute caffeine (CAF: 375 mg, ≈4.8 mg/kg body mass) effects on energy expenditure (EE) and substrate kinetics during high-intensity interval exercise in individuals with high (HBAT) versus low (LBAT) brown adipose tissue activity using time-trend polynomial modeling. **Methods**: This is a randomized, double-blind crossover study in which 35 highly-trained males [HBAT-CAF, HBAT-PLA (Placebo), LBAT-CAF, LBAT-PLA] performed 30-min treadmill HIIE. Infrared thermography (IRT) assessed BAT activity by measuring supraclavicular skin temperature (SST). Breath-by-breath ergospirometry measured EE (kcal/min) and carbohydrate (CHO), lipid (LIP), and protein (PTN) oxidation. We applied second- and third-order polynomial regression models to depict the temporal trajectories of metabolic responses. **Results**: HBAT groups showed 25% higher sustained EE versus LBAT (*p* < 0.001), amplified by CAF. CHO oxidation exhibited biphasic kinetics: HBAT had 40% higher initial rates (0.75 ± 0.05 vs. 0.45 ± 0.04 g/min; *p* < 0.001) with accelerated decline (k = −0.21 vs. −0.15/min; *p* = 0.01). LIP oxidation peaked later in LBAT (40 vs. 20 min in HBAT), with CAF increasing oxidation by 18% in LBAT (*p* = 0.01). HBAT-CAF uniquely showed transient PTN catabolism (peak: 0.045 g/min at 10 min; k = −0.0033/min; *p* < 0.001). **Conclusions**: BAT status determines EE magnitude and substrate-specific kinetic patterns, while CAF exerts divergent modulation, potentiating early glycogenolysis in HBAT and lipolysis in LBAT. The HBAT-CAF synergy triggers acute proteolysis, revealing BAT-mediated metabolic switching.

## 1. Introduction

Emerging research highlights brown adipose tissue (BAT) as a metabolically active organ possessing both thermogenic and endocrine functions. BAT secretes signaling molecules termed “batokines,” which modulate systemic metabolism by influencing skeletal muscle, liver, pancreatic, and neural activity [1,2]. These interactions enhance glucose and fatty acid uptake in muscle, suppress hepatic lipogenesis, improve beta-cell function, and mitigate cardiovascular strain, collectively promoting overall metabolic health and aiding in management of dyslipidemia and hypertension [2,3,4]. Despite advances in understanding metabolic pathways, gaps persist in translating the therapeutic potential of BAT. Studies suggest that activating as little as 50 g of BAT may elevate metabolic rate by up to 25%, offering a promising strategy for metabolic modulation [5].

Brown adipose tissue was once thought to be exclusive to infants, but is now known to persist in adults, as demonstrated by 18F-fludeoxyglucose positron emission tomography (18F-FDG PET/CT) imaging [6]. While 18F-FDG PET/CT remains the gold standard for identifying metabolically active BAT through glucose analog uptake, its clinical utility is limited by radiation exposure and cost [7]. On the other hand, Infrared thermography (IRT) presents a non-invasive, low-cost alternative by measuring skin temperature in BAT-rich regions, such as the supraclavicular area, where superficial deposits are abundant [8,9].

IRT has appeared as a promising non-invasive method for assessing BAT activity through measuring the supraclavicular skin temperature (SST) [10,11,12]. IRT quantifies heat emission (a proxy for BAT thermogenesis) before, during, or after cold or pharmacological stimulation, and agreement with PET/CT confirms its reliability [10,11]. A systematic review indicates that SST is increasingly recognized as a surrogate marker for BAT volume and function, with 18 out of 24 studies linking SST changes to human BAT activity [12]. This approach leverages BAT’s inherent capacity for heat generation via non-shivering thermogenesis, a process detectable as localized surface warming [6,12,13]. Specifically, SST positively correlates with BAT activity, especially during cold exposure where heightened metabolic activity elevates local skin temperature [12,14,15]. IRT crucially offers a comprehensive, cost-effective alternative to traditional techniques like 18F-FDG PET, which estimates BAT activation via glucose uptake [14]. Its non-invasive nature further enables repeated assessments, providing dynamic insights into BAT behavior across physiological and pathological states [13]. Consequently, IRT stands as a valuable tool for evaluating BAT’s role in metabolism, allowing clinicians and researchers to monitor changes without invasive procedures [14,15].

BAT thermogenesis is driven by mitochondrial uncoupling protein 1 (UCP-1), which dissipates energy as heat [16]. Ergogenic compounds like capsaicin analogs and capsinoids enhance UCP-1 activity, boosting energy expenditure [17,18]. Similarly, caffeine (CAF) stimulates BAT via β-adrenergic pathways, upregulating UCP-1, proliferator-activated receptor γ, and peroxisome proliferator-activated receptor-gamma coactivator to promote mitochondrial biogenesis and thermogenesis [5,19]. These mechanisms align with the observed effects of caffeine on weight loss and metabolic rate increase [16].

In addition to caffeine, exercise stimulates energy expenditure through sympathetic nervous system (SNS) activation, which enhances skeletal muscle metabolism and BAT thermogenesis [4,18]. SNS-mediated norepinephrine release binds β-adrenergic receptors, upregulating UCP-1 and peroxisome proliferator-activated receptor-gamma coactivator via the p38 mitogen-activated protein kinase pathway [20]. Exercise also elevates adipocyte triglyceride lipase and hormone-sensitive lipase activity, promoting lipid mobilization and white adipose tissue browning [21]. Concurrently, exercise-induced irisin secretion stimulates adipose tissue thermogenesis through mammalian p38 mitogen-activated protein kinase signaling, further linking physical activity to metabolic health [20]. Despite growing interest in BAT activation strategies, limited research explores the acute effects of exercise on BAT-mediated energy expenditure (EE), particularly in combination with ergogenic supplements. This study investigates whether individuals with HBAT (high) versus LBAT (low brown adipose tissue) activity exhibit differential energy expenditure during and after 28 min of high-intensity interval exercise (HIIE), with or without CAF supplementation. We hypothesized that individuals’ HBAT-CAF would exhibit greater EE and lipid (LIP) catabolism compared to HBAT-PLA (placebo), LBAT-CAF, and LBAT-PLA.

## 2. Materials and Methods

### 2.1. Experimental Approach

This study employs a quasi-experimental design because participants were pre-classified into HBAT and LBAT groups based on their basal brown adipose tissue activity, which is an intrinsic variable not subject to randomization. Although the intervention (CAF/PLA) was randomized and controlled, the primary exposure (BAT status) was determined observationally. This protocol was previously approved by ethical committee of the Federal University of Juiz de Fora (CAAE: 35605220.1.0000.5147, register number: 4.366.750) and registered in the Brazilian clinical trials network—REBEC. On the first day of data collection, participants were categorized into either the HBAT or LBAT group based on the thermographic protocol established by Nirengi et al. [14]. On the next day, a researcher who was not involved in data collection randomly (by coin toss) assigned participants to either the CAF or PLA conditions. Participants then underwent an HIIE protocol lasting 30 min, followed by 30 min of passive recovery. Then, the conditions were reversed (crossover design) on the third day (seven days after the second session), ensuring that all participants experienced both CAF and PLA conditions. This approach ensured that the study was conducted in a double-blind manner, minimizing bias. Participants’ physiological responses were monitored throughout both exercise and recovery periods using an ergospirometry protocol to collect relevant data. Figure 1 shows the timeline procedures.

### 2.2. Participants

The researchers established the following inclusion criteria to achieve the objectives of the present study: (a) male; (b) aged 18 years or older; (c) physically active, with a training frequency of at least four times per week; (d) free from any injuries that could impair test performance; and (e) not using any stimulant or vasodilator medications. From the initial pool of participants, those who (a) did not complete all stages of the study for any reason or (b) had errors during the collection or processing of ergospirometry data were excluded.

The minimum sample size estimate was based on two previous studies with similar research subjects [11,22]. EE was applied as the primary outcome for sample size estimation. A power analysis was conducted with the following parameters: α (significance level) set at 0.05 (two-tailed), 95% confidence interval (95% CI), power (1−β) of 0.8, and a Cohen’s d effect size of 0.2. Based on these criteria, a minimum of 13 participants per group was required to detect a clinically meaningful difference of 100 kcal between groups and conditions (i.e., HBAT-CAF, LBAT-CAF, HBAT-PLA, LBAT-PLA). Therefore, a total of 40 male highly trained athletes were invited to participate in the study, of whom 37 began the protocol and 35 completed it. The groups exhibited the following anthropometric characteristics: (a) HBAT (n = 15): 26.5 ± 4.3 years; 1.7 ± 0.1 m; 77.4 ± 7.2 kg; 25.5 ± 1.8 kg/m^2^; 15.7 ± 2.0% body fat; and (b) LBAT (n = 20): 27.0 ± 4.1 years; 1.7 ± 0.1 m; 79.0 ± 8.1 kg; 26.0 ± 1.7 kg/m^2^; 15.9 ± 4.5% body fat. No significant differences were observed between the two groups for these variables (*p* ≥ 0.05). Of the total, 94.3% (n = 33) consumed at least 4 cups of coffee (75–100 mL) per day. Figure 2 shows the participants’ allocations, completed for group and condition.

### 2.3. BAT Protocol and Classification

The classification of participants regarding BAT activity was estimated using a thermographic protocol validated by positron emission tomography—PET-CT (gold-standard) [14]. In this protocol, participants (wearing only shorts) remained seated in a climate-controlled room at 19 °C and 48% humidity for 60 min. After this period, two thermal images of the upper trunk region were captured following the thermographic imaging protocol proposed by Moreira et al. [23]. The highest-quality image was selected for analysis. BAT activity was calculated based on the temperature difference between the mean of the supraclavicular region and a reference external region (control). Participants with a temperature difference of ≥1.03 °C were classified into the HBAT group. This cut-off demonstrated a specificity of 84.6%, sensitivity of 85.7%, and accuracy of 85.4% [14].

A thermal imaging camera (FLIR T335^®^, FLIR Systems, Taby, Sweden) with a resolution of 320 × 240 pixels, a spectral range of 7.5 to 13.0 µm, an image frequency of 30 Hz, a thermal sensitivity of 50 mK at 30 °C, and an accuracy of ±2% was used for registering the thermograms. The emissivity values were set at 0.98, corresponding to the emissivity of human skin. A thermographic image (thermogram) was captured, including the anterior cervical and supraclavicular regions, at 0, 10, 20, 30, 40, 50, and 60 min after starting the exercise protocol. Figure 3 illustrates an example of the regions of interest selected for the BAT activity estimation by infrared radiation.

FLIR Tool + analysis software (FLIR, Taby, Sweden) was used to obtain Tsk data from the recorded thermograms. With this software and following the procedure described by Yoneshiro et al. [22], regions of interest (ROIs) were delimited by selecting two 21 × 21 pixel circles, one above the left median clavicular region (1) and the other in the upper thorax (2). The ROIs were selected on the left side of the subject because previous studies indicate that the pulmonary vascular bundle influences the results on the right side of the subject [22].

### 2.4. Pre-Exercise Breakfast and Supplement

Data collection was conducted always in the early morning to minimize variability and control for potential circadian influences. The laboratory was maintained at a controlled temperature of 21 °C and a relative humidity of 55% to mitigate potential biases due to ambient temperature on BAT activity and HIIE performance [24]. Participants were instructed to arrive at the laboratory at 07:00 a.m., where they were provided with a standardized breakfast totaling approximately 320 kcal. The meal included a medium banana (≈90 kcal), two slices of whole-wheat bread (140 kcal) with peanut butter (≈90 kcal), and lemon-flavored water (0 kcal). This protocol ensured that all participants began the study under consistent nutritional and physiological conditions. Participants were also administered a capsule at breakfast containing either (a) CAF (375 mg, ≈4.8 mg/kg body mass) or (b) PLA (maltodextrin), depending on their assigned experimental condition. They were randomly assigned to receive either CAF or PLA capsules, which were indistinguishable in smell, color, and flavor. Athletes who initially received CAF were subsequently administered PLA in the data collection (and vice versa) in order to adhere to the crossover design. A researcher uninvolved in data collection prepared and distributed the capsules, ensuring adherence to the double-blind protocol by preventing communication of group assignments to both participants and the research team conducting assessments. No participant experienced any side effects associated with CAF consumption.

### 2.5. High-Intensity Interval Exercise, Recovery Protocol, and Ergospirometric Measurements

A portable breath-by-breath gas analyzer (Metalyzer 3BR3^®^, Cortex, Leipzig, Germany) was used to collect metabolic data at seven time points: 0, 10, 20, 30, 40, 50, and 60 min. Measurements from 0 to 30 min corresponded to the effort phase, while the subsequent 30 min represented the recovery phase. Energy expenditure was measured in 10-min intervals during the test to capture physiological changes coinciding with the absorption of the capsule contents (CAF or PLA). This interval aligns with caffeine pharmacokinetics, showing peak plasma concentrations occurring 30–60 min post-ingestion [25,26] and evidence that ergogenic effects can emerge within 30 min, depending on capsule type [26]. EE data were presented in kcal/min and macronutrient catabolism in g/min, according the recommendations by Jeukendrup and Wallis [27] and Gonzalez et al. [28]. The device was calibrated prior to each session in accordance with the manufacturer’s specifications.

One hour after consuming breakfast and ingesting the randomized capsules (CAF or PLA), participants underwent a modified high-intensity interval exercise (HIIE) treadmill protocol as per Tjønna et al. [29]. The regimen consisted of four 4-min sprint intervals at 90–95% of maximum heart rate (HRmax), each followed by a 3-min active recovery phase at 60–70% of HRmax. Although the original protocol lasted 28 min, the final recovery phase in this study was extended by 2 min to standardize the total exercise duration to 30 min. HRmax was calculated using the equation proposed by Tanaka et al. [30]: *HRmax = 208 − (0.7 × age)*.

Following the HIIE session, participants rested in the supine position for 30 min while recovery metabolism was assessed via indirect calorimetry. Oxygen consumption and carbon dioxide production were continuously monitored throughout the experiment to estimate total energy expenditure (EE; Kcal) and substrate utilization rates (g/min) for carbohydrates (CHOs), lipids (LIPs), and proteins (PTNs). CHO and FAT oxidation rates were derived using Frayn’s stoichiometric equations as in Alcantara et al. [31]. Protein (PTN) catabolism was estimated using the manufacturer’s protocol (Metasoft^®^ Studio 5.5.1, Cortex, Leipzig, Germany), which implements Weir’s stoichiometric equations based on urinary urea nitrogen excretion [32]. This approach is consistent with established methodologies for indirect protein oxidation assessment [33].

### 2.6. Data Analysis

Data were initially organized and managed in spreadsheets using Microsoft Excel 2024 (Microsoft Corporation, Redmond, WA, USA). The 95% confidence interval (95% CI) was calculated using the sample mean, standard deviation, degrees of freedom, and sample size. We applied second-and third-order polynomial regression models to depict the temporal trajectories of metabolic responses (e.g., EE, CHO, LIP, PTN). Model selection was performed via the Akaike Information Criterion (AIC), with preference given to the polynomial order minimizing AIC [34]. Curve fitting was implemented using least-squares algorithms in Python 3.10 (SciPy v1.13.0 and Matplotlib v3.8.0 libraries). All analyses were performed with statistical significance defined a priori as *p* ≤ 0.05.

## 3. Results

Regarding the models used for temporal analysis to characterize the kinetic patterns of energy expenditure and substrate oxidation, the temporal trajectories revealed previously unidentified nonlinear patterns in metabolic responses between the experimental groups. Figure 4 illustrates the time-trend analysis for EE.

According to Figure 4, the HBAT groups maintained significantly elevated EE throughout the protocol (+25% vs. LBAT; *p* < 0.001). The HBAT-CAF group demonstrated the highest EE values at all time points (4.0 ± 0.2 kcal/min in min1 vs. 3.1 ± 0.3 in LBAT-PLA; *p* = 0.003). Third-order polynomial models revealed a characteristic linear decline pattern (R^2^ = 0.95), with no significant differences in decay rates between groups (k = −0.15 ± 0.02/min; *p* = 0.12). Figure 5 shows the time-trend analysis for CHO.

Figure 5 shows a biphasic pattern of CHO utilization. The HBAT groups exhibited 40% higher initial oxidation rates (0.75 ± 0.05 g/min vs. 0.45 ± 0.04 g/min in LBAT at min1; *p* < 0.001), with significantly steeper decline kinetics during the glycogen-dependent phase (mins 1–3; k_hmat_ = −0.21 ± 0.03/min vs. k_lmat_ = −0.15 ± 0.02/min; *p* = 0.01). Figure 6 shows the time-trend analysis for LIP catabolism.

Contrary to CHO patterns, lipid utilization (Figure 6) showed late maximal activation. Peak oxidation occurred at 40 min in LBAT-CAF (0.12 ± 0.01 g/min) versus 30 min in LBAT-PLA (0.10 ± 0.01 g/min). The HBAT groups demonstrated early maximal activation (20 min) with 15% higher maximal oxidation (*p* = 0.02). Caffeine increased lipid oxidation by 18% in LBAT (*p* = 0.01), but not in HBAT (*p* = 0.32). Figure 7 shows the time-trend analysis for LIP catabolism.

The HBAT-CAF group exhibited a unique kinetic profile (Figure 7), characterized by a transient peak of protein catabolism (0.045 ± 0.005 g/min at 10 min) followed by a rapid decline (k = −0.0033/min; *p* < 0.001). This pattern was absent in the other groups, which maintained stable protein utilization (0.033 ± 0.004 g/min; *p* = 0.87 for time effect). Table 1 summarizes the main metabolic kinetics for EE and macronutrients.

The summary of metabolic kinetics, highlighting the distinct advantages associated with high body adiposity training (HBAT) and the influence of caffeine on macronutrient utilization and energy expenditure. Energy expenditure was significantly elevated by 25%, with a strong association with HBAT, and followed a linear decline over time. Carbohydrate metabolism showed a marked increase of 40%, characterized by a sharp initial phase in HBAT and a biphasic decay pattern. Lipid oxidation increased by 15%, with a notable advantage observed in low body adiposity training (LBAT), peaking in the later stages of the metabolic response. Protein kinetics revealed a transient peak with a rapid initial decline, suggesting a specific and transient effect of HBAT. Notably, the lipid advantage pertained only to the peak of oxidation and was not maintained over time.

## 4. Discussion

While BAT is a key regulator of energy homeostasis, the combined effects of thermogenic compounds and exercise on BAT-mediated kinetic patterns remain underexplored in humans [35]. This study is the first to investigate the acute interaction between HIIE and CAF in individuals stratified in HBAT and LBAT using temporal trend kinetic modeling. Our hypothesis was partially supported, as the HBAT-CAF group exhibited significantly higher and sustained EE during and post-exercise. BAT activity determined substrate-specific kinetic patterns: HBAT groups showed biphasic carbohydrate oxidation with accelerated initial-phase decline, LBAT demonstrated delayed lipid oxidation peaks, and HBAT-CAF uniquely triggered transient proteolysis. These findings reveal that the HIIE-CAF combination amplifies EE through BAT-mediated metabolic switching (a process where caffeine potentiates early glycogenolysis) [36]. Prior studies have confirmed the capacity of caffeine to stimulate BAT thermogenesis [5,11], but the substrate kinetic modulation observed herein provides new mechanistic insights into exercise-thermogenic synergy.

Our results bridge this gap, revealing that CAF supplementation (≈4.8 mg/kg) amplified EE across all groups, irrespective of BAT activation status, consistent with previous studies by Pérez et al. [11]. CAF likely potentiates HIIE-induced SNS activation, as it stimulates β-adrenergic receptors to enhance metabolic and cardiovascular activity [5]. This synergy may explain the pronounced EE in HBAT-CAF, where elevated norepinephrine levels during HIIE [36,37] and post-exercise cortisol release [38] likely amplified thermogenesis. CAF has been shown to synergistically enhance BAT activity in previous studies [16,19,22]. CAF increases the expression of essential thermogenic genes, such as UCP1, thereby promoting BAT functional capacity [5,39]. Specifically, the caffeine-induced increase in cAMP levels leads to enhanced sympathetic drive for BAT thermogenesis [19,39]. Peak EE notably occurred between 10 and 30 min of HIIE, aligning with studies linking exercise intensity to catecholamine surges [38,40]. While HIIE predominantly relies on anaerobic glycolysis, aerobic pathways contribute to sustained lipolysis and EE [41,42]. Despite these effects, BAT activation did not modulate substrate oxidation, contrasting with the study by Mekonen et al. [43], who identified exercise-induced proteolysis as a minor but measurable energy source.

The effects of HIIE on BAT metabolism remain poorly understood, with conflicting evidence in the literature [44]. For instance, Li et al. [45] conducted a comparative magnetic resonance image study involving 10 athletes performing HIIE and 20 individuals engaging in moderate-intensity exercise, revealing significantly higher BAT volume in the HIIE group. In fact, HIIE can induce greater norepinephrine secretion compared to aerobic exercise [1], which may enhance BAT activation through stimulation of thermogenesis and lipolysis pathways [2,3]. HIIE has specifically been shown to increase mitochondrial density and elevate the expression of thermogenic markers, correlating with the remodeling of metabolic processes [4,5]. In a comparative MRI analysis, mice undergoing HIIE displayed significant improvements in BAT activity, indicating that HIIE may increase the thermogenic capacity [46]. Conversely, Motiani et al. [47] reported only partial alignment with our findings: while individuals with high HBAT exhibited a higher EE than those LBAT, the HIIE protocol was associated with reduced insulin signaling in BAT, suggesting that HIIE may not sufficiently stimulate BAT activation. Methodological differences such as the use of repeated HIIE sessions and the absence of dietary supplementation in the protocol by Motiani et al. may notably account for discrepancies compared to our experimental design. Further complicating the evidence, Scheel et al. [44] underscore in their review that exercise-induced BAT activation remains understudied, with limited human trials demonstrating positive outcomes and a predominance of preclinical animal studies dominating the field. This collective ambiguity highlights a critical knowledge gap, emphasizing the need for rigorous, well-controlled human studies to elucidate the role of HIIE in modulating BAT metabolism. Although not the focus of this investigation, previous studies have indicated that HIIE combined with caloric restriction can promote remodeling of glucose and lipid metabolism [48,49], while simultaneously enhancing mitochondrial function in brown and beige adipocytes [48].

The relationship between HIIE and PTN catabolism has been explored in some investigations [50,51,52]. Recently, Mesquita et al. [52] demonstrated that while 7 weeks of resistance training promoted significant anabolic adaptation in the quadriceps, the same duration of HIIE triggered marked catabolic effects in this muscle group. Muscle biopsies of the *vastus lateralis* revealed elevated expression of the Muscle Atrophy F-box/Atrogin-1 marker, a key regulator of proteolysis mediated by the ubiquitin–proteasome system. Parallel findings were reported by Haun et al. [51], who observed increased catabolic signaling following a short-term, intensive HIIE protocol (3 consecutive days, 22.5 min/session, 45 s work/rest intervals). These results highlight a potential divergence in muscle metabolic responses to resistance training versus HIIE. Future research should investigate the chronic combined effects of HIIE and CAF, particularly in HBAT individuals, as this combination may transiently amplify metabolic demands. In addition, targeted protein supplementation strategies could help attenuate nitrogen imbalance during such interventions without impairing exercise performance [53].

Despite the innovative results, our protocol presents limitations, which include the indirect assessment of supraclavicular BAT via thermography, which may underestimate heat emission in individuals with low BAT volume [14]. The 85% accuracy in BAT classification also introduces risks of misgrouping. Critically, a cost–benefit analysis underscores thermography’s viability for studying BAT activity compared to 18F-FDG PET/CT. The latter involves ionizing radiation (limiting repeat measurements per participant and preventing its use in vulnerable populations) [54], which is costly, invasive, and non-portable. Despite its lower accuracy, the thermographic protocol offers distinct logistical and safety advantages over the gold-standard method.

The process of daily measurement of energy expenditure and macronutrient catabolism constitutes another limitation of the study, subject to the restrictions of the processing software used. Furthermore, the CAF dosage was not tailored to individual participants, leading to variability in the relative amount intake. While this study focused on physically active male individuals, future research should prioritize populations with obesity or overweightness to better align findings with weight loss applications. It will also be interesting if future protocols measure people of different ages and women. Investigations could explore how moderate-intensity exercise, CAF intake, and BAT activity interact across diverse demographics, including females, sedentary populations, and individuals with metabolic imbalances, in order to enhance translational relevance. Such studies would clarify whether these interventions elicit universal or context-dependent metabolic responses. Furthermore, longitudinal research is critical to evaluate the sustained effects of combining exercise, supplementation, and BAT activation, as short-term outcomes may not reflect enduring metabolic adaptations.

## 5. Conclusions

Our findings demonstrate that caffeine supplementation acutely increases EE during HIIE, with a more pronounced sustained effect in HBAT individuals. BAT activity modulates substrate-specific kinetic patterns, inducing biphasic carbohydrate oxidation with an accelerated decline in the early phase, shifting the lipid oxidation peak to late phases in LBAT individuals, and triggering transient proteolysis exclusively in the HBAT-CAF condition. Caffeine exerts divergent modulation, potentiating early glycogenolysis in HBAT and lipolysis in LBAT. These results suggest that the combination of HIIE and caffeine optimizes EE through BAT-mediated metabolic switching. Future studies should investigate long-term adaptations and mechanisms underlying the kinetic modulation of substrates by BAT.

## Figures and Tables

**Figure 1 metabolites-15-00517-f001:**
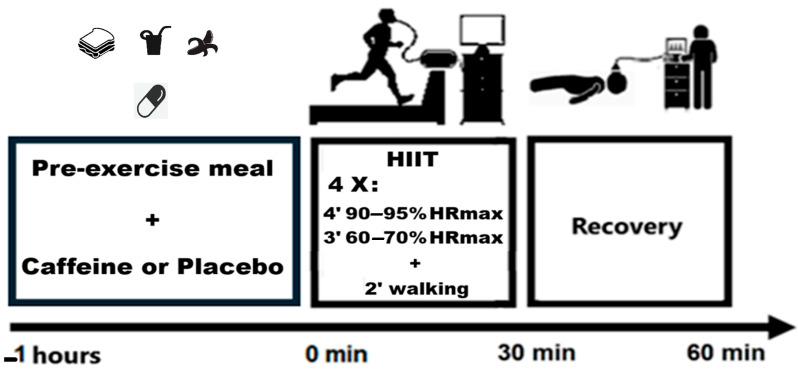
Timeline of experimental procedures. HIIE—high intensity interval exercise, HRmax—maximum heartrate.

**Figure 2 metabolites-15-00517-f002:**
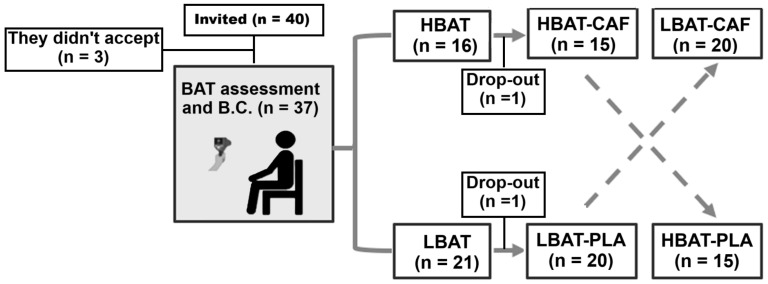
Participant selection, group classification, and condition trials. BAT—brown adipose tissue, HBAT—high brown adipose tissue activity, LBAT—low brown adipose tissue activity, CAF—caffeine, PLA—placebo.

**Figure 3 metabolites-15-00517-f003:**
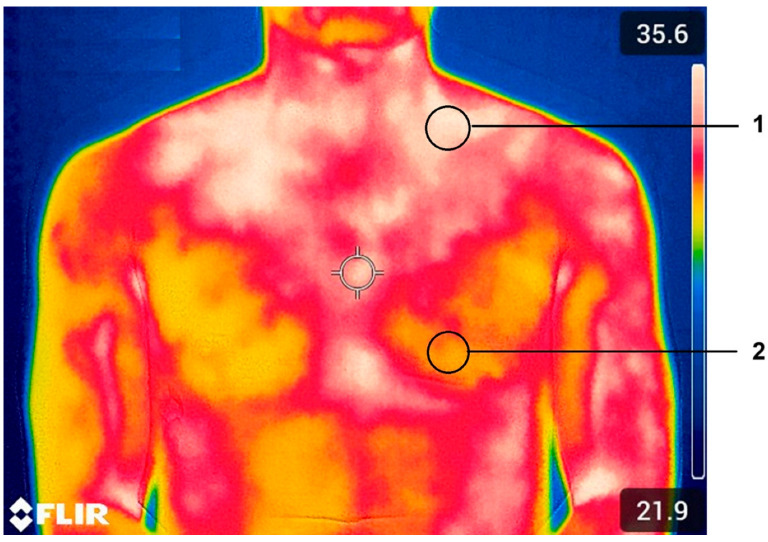
Example of BAT activity estimation. 1: supraclavicular region of interest and 2: external region of interest.

**Figure 4 metabolites-15-00517-f004:**
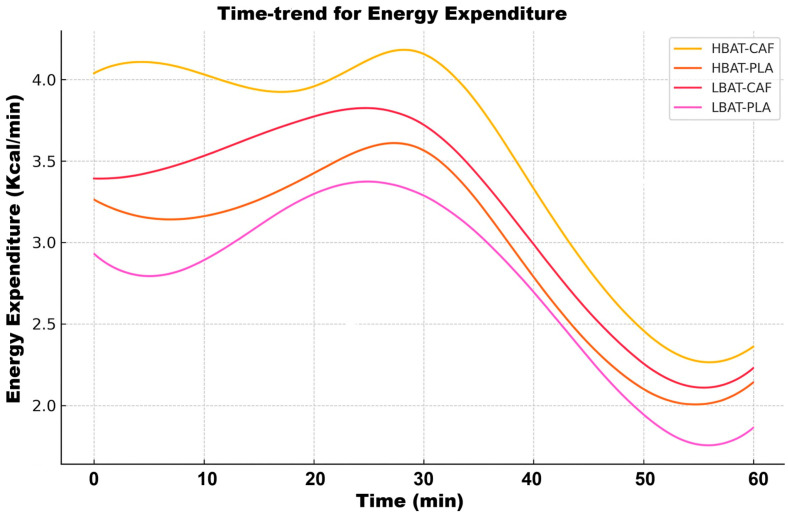
Time-trend analysis for energy expenditure for all groups and conditions. HBAT-CAF (yellow line), HBAT-PLA (brown line), LBAT-CAF (orange line), and LBAT-PLA (pink line).

**Figure 5 metabolites-15-00517-f005:**
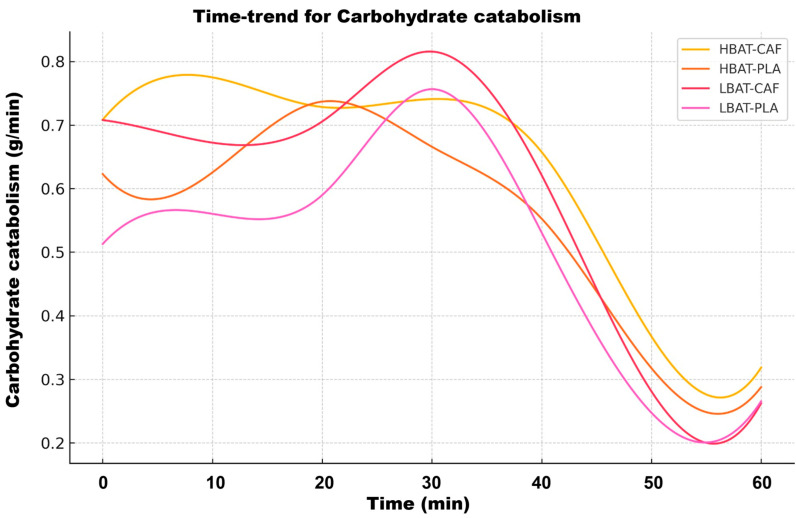
Time-trend analysis for carbohydrate catabolism for all groups and conditions. HBAT-CAF (yellow line), HBAT-PLA (brown line), LBAT-CAF (orange line), and LBAT-PLA (pink line).

**Figure 6 metabolites-15-00517-f006:**
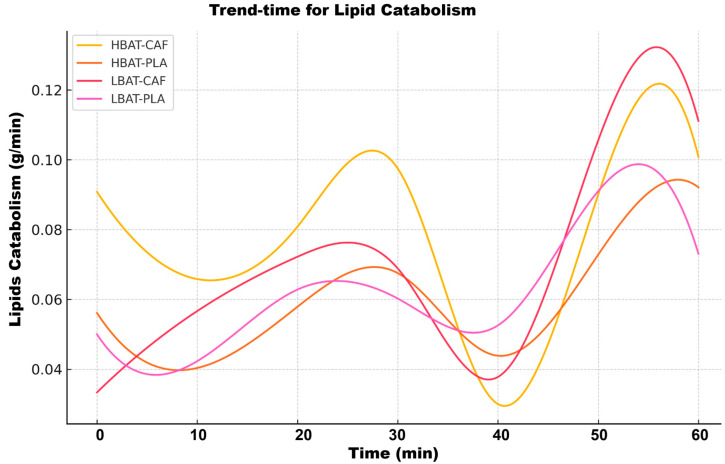
Time-trend analysis for lipid catabolism for all groups and conditions. HBAT-CAF (yellow line), HBAT-PLA (brown line), LBAT-CAF (orange line), and LBAT-PLA (pink line).

**Figure 7 metabolites-15-00517-f007:**
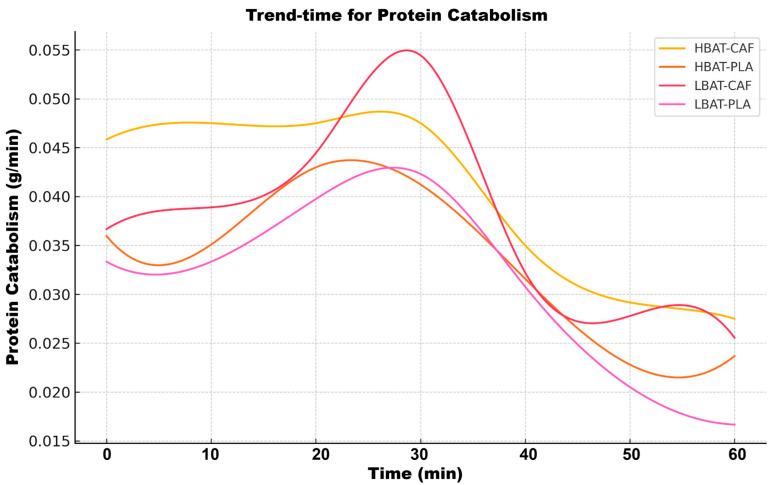
Time-trend analysis for protein catabolism for all groups and conditions. HBAT-CAF (yellow line), HBAT-PLA (brown line), LBAT-CAF (orange line), and LBAT-PLA (pink line).

**Table 1 metabolites-15-00517-t001:** Comparative summary of metabolic kinetics.

Measure	Advantage for HBAT	Effect of Caffeine	Kinetic Pattern
**Energy Expenditure**	↑ 25%	Strong for HBAT	Linear Decline
**Carbohydrates**	↑ 40%	Sharp initial phase in HBAT	Biphasic Decay
**Lipids**	↑ 15% *	Strong in LBAT	Late Peak
**Proteins**	Transitory peak	Specific effect in HBAT	Initial Rapid Decline

* Advantage in maximum oxidation (not sustained).

## Data Availability

The database for this study will be made available on demand; just send an email to ciro.brito@ufjf.br.

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
