# Peer review of "Thermogenic Activation of Adipose Tissue by Caffeine During Strenuous Exercising and Recovery: A Double-Blind Crossover Study"

_metabolites, 2025, doi:10.3390/metabo15080517_

Round 1
Reviewer 1 Report
Comments and Suggestions for Authors
The authors employed A quasi-experimental crossover intervention design was used to study how immediate caffeine (CAF) consumption, versus placebo (PLA), affects energetic activation (using infrared thermography) in trained individuals with high (HBAT) and low (LBAT) proportion of brown fat, after high-intensity interval (HIIE, treadmill, 90–95% of HRmax / 30 min; passive recovery, 30 min) exercise. Total energy expenditure (EE) and use of ergogenic substrates [Carbs (CHO), fat, protein (PTN)] were measured. The authors observed that CAF-supplemented groups exhibited more EE during exercise and recovery than non-supplemented ones, upon their BAT status (HBAT-CAF > LBAT-CAF >HBAT-PLA, LBAT-PLA) while differences in substrate utilization were treatment (e.g. high PTN oxidation rate in HBAT-CAF group)- and time-dependent. Despite the study's small scale, it provided enough concrete evidence to achieve its goal. However, several aspects of its form and content require modification/improvement to highlight its unique contribution to the field. Here are some observations:
General. A) The English translation could use some work. I recommend having a native English speaker or a professional translation service review the next draft. B) Too many abbreviations are used throughout; define each abbreviation the first time it appears and remove any rarely used ones. C) Italicize words when needed [e.g., scientific names, Latin terms (e.g. et al., in vivo], even in references.
Title. Quite long. Suggestion: Thermogenic activation of adipose tissue by caffeine during strenuous exercising and recovery.
Abstract. A) Once the results have been modified according to the suggestions described below, it is suggested to reconstruct them quantitatively (including p-values) rather than qualitatively. B) Descriptions should reflect the independent effects of both caffeine supplementation and body`s BAT content.
Introduction. A) 1st paragraph- Given the participants' high fitness level and optimal body fat percentage, any mention to “obesity” should be removed from the manuscript or at least use “excess weight” (participant`s IMC ~ 24-26). Do from this section onwards. B) 2nd paragraph - The authors should elaborate on the justification for infrared thermography as a suitable method for detecting brown fat deposits.
Methods. A) Since the authors tracked certain variables over time (Figure 1, Tables 1-2), it's best to present the data at a specific point in time and create time-trend kinetic graphs showing how each variable changed differently upon experimental groups; Induction based on 2nd – 3rd order is more informative than raw data.
Results/discussion. A) A more inductive and integrative rather than descriptive discussion of all results is strongly recommended. It's best to compare this study to similar ones, highlighting what's new and different about it.
Tables. They should be transformed to time-trend kinetic graphs, analyzing the numerical data derived from their equations to highlight trends across treatments during the whole period instead of at a specific point in time.
Figures. A) Do not forget to provide them with enough resolution (≥300 dpi).
Conclusion. Change if necessary, according to new suggestions
References. OK.
Comments on the Quality of English LanguageThe English could be improved to more clearly express the research
Author Response
Reviewer 1
The authors employed A quasi-experimental crossover intervention design was used to study how immediate caffeine (CAF) consumption, versus placebo (PLA), affects energetic activation (using infrared thermography) in trained individuals with high (HBAT) and low (LBAT) proportion of brown fat, after high-intensity interval (HIIE, treadmill, 90–95% of HRmax / 30 min; passive recovery, 30 min) exercise. Total energy expenditure (EE) and use of ergogenic substrates [Carbs (CHO), fat, protein (PTN)] were measured. The authors observed that CAF-supplemented groups exhibited more EE during exercise and recovery than non-supplemented ones, upon their BAT status (HBAT-CAF > LBAT-CAF >HBAT-PLA, LBAT-PLA) while differences in substrate utilization were treatment (e.g. high PTN oxidation rate in HBAT-CAF group)- and time-dependent. Despite the study's small scale, it provided enough concrete evidence to achieve its goal. However, several aspects of its form and content require modification/improvement to highlight its unique contribution to the field. Here are some observations:
We appreciate the time you dedicated to our manuscript, your evaluation was thorough and positive, indicating crucial points for improvement.
General. A) The English translation could use some work. I recommend having a native English speaker or a professional translation service review the next draft. B) Too many abbreviations are used throughout; define each abbreviation the first time it appears and remove any rarely used ones. C) Italicize words when needed [e.g., scientific names, Latin terms (e.g. et al., in vivo], even in references.
Thank you for your comments. None of the authors are native English speakers, so the text was reviewed by a professional proofread. I will pass on the comments to them before submitting the next version. We also observed that there were acronyms that were not explained in the summary such as CAF. We have removed all acronyms that appear less than 3 times in the text. We reviewed all the text and inserted italics where necessary.
Title. Quite long. Suggestion: Thermogenic activation of adipose tissue by caffeine during strenuous exercising and recovery.
Thanks for the suggestion, we have changed the title. However, we inserted your title with a small modification because reviewer 2 requested that our study follows CONSORT for crossover studies.
Abstract. A) Once the results have been modified according to the suggestions described below, it is suggested to reconstruct them quantitatively (including p-values) rather than qualitatively. B) Descriptions should reflect the independent effects of both caffeine supplementation and body`s BAT content.
The Abstract was completely rewritten due to the need to adjust to the new results.
Introduction. A) 1st paragraph- Given the participants' high fitness level and optimal body fat percentage, any mention to “obesity” should be removed from the manuscript or at least use “excess weight” (participant`s IMC ~ 24-26). Do from this section onwards. B) 2nd paragraph - The authors should elaborate on the justification for infrared thermography as a suitable method for detecting brown fat deposits.
Thank you for this observation. Indeed, all reviewers noted that discussing obesity while studying athletes was inconsistent. We have modified the text, removed the references related to obesity, and also added a paragraph reporting the importance of thermography for estimating BAT activity. As suggested by another reviewer, we place targeted research on obesity and overweight as a focus of future studies.
Methods. A) Since the authors tracked certain variables over time (Figure 1, Tables 1-2), it's best to present the data at a specific point in time and create time-trend kinetic graphs showing how each variable changed differently upon experimental groups; Induction based on 2nd – 3rd order is more informative than raw data.
We sincerely appreciate the reviewer's suggestion regarding the temporal analysis of our data. In response, we implemented time-trend kinetic modeling, where we performed polynomial regression analysis (2nd and 3rd order) for all metabolic variables (EE, CHO, FAT, PTN) across exercise and recovery. Using the Python libraries matplotlib and scipy, we generated four kinetic graphs (i.e.; EE, CHO, FAT, PTN) showing specific response trajectories for each group.
Results/discussion. A) A more inductive and integrative rather than descriptive discussion of all results is strongly recommended. It's best to compare this study to similar ones, highlighting what's new and different about it.
We thank the reviewer for the suggestion; we kept some paragraphs and inserted others, where we tried to make a more integrated analysis of the results.
Tables. They should be transformed to time-trend kinetic graphs, analyzing the numerical data derived from their equations to highlight trends across treatments during the whole period instead of at a specific point in time.
We thank the reviewer for suggesting this analysis method to me; the results really were more presentable. As previously reported, we made the figures for the measurements of CHO, LIP and PTN, as suggested by the reviewer. We have inserted a new Table, which generalizes the main results of the models.
Figures. A) Do not forget to provide them with enough resolution (≥300 dpi).
The new 4 Figures were sent in this high-resolution version.
Conclusion. Change if necessary, according to new suggestions.
We kept the structure of the previous version, but we had to modify and rewrite it according to the new results.
References. OK.
We have added new references based on your suggestions.
Reviewer 2 Report
Comments and Suggestions for Authors
The aim of this study was to examine the effect of high intensity physical exercise and caffeine supplementation on energy expenditure in trained young healthy males categorized as high or low brown adipose tissue activity by thermography. The results show that energy expenditure during high intensity exercise was higher in HBAT thatn in LBAT persons and caffeine increased energy expenditure in both groups. In addition, carbohydrate, lipid and protein metabolism were estimated by respirometry.
- The aim of the study specified in the Abstract is not consistent with the manuscript title since the effect of caffeine is not mentioned.
- The study was performed in males only. Could the results be extrapolated to females?
- The results are partially expectable since the effect of BAT activity on energy expenditure is well established.
Author Response
Reviewer 2
The aim of this study was to examine the effect of high intensity physical exercise and caffeine supplementation on energy expenditure in trained young healthy males categorized as high or low brown adipose tissue activity by thermography. The results show that energy expenditure during high intensity exercise was higher in HBAT thatn in LBAT persons and caffeine increased energy expenditure in both groups. In addition, carbohydrate, lipid and protein metabolism were estimated by respirometry.
We sincerely appreciate the time and effort you dedicated to reviewing our manuscript.
The aim of the study specified in the Abstract is not consistent with the manuscript title since the effect of caffeine is not mentioned.
We are very grateful to the reviewer for the observation. This was also pointed out by other reviewers. We rewrote the abstract to remove this inconsistency.
The study was performed in males only. Could the results be extrapolated to females?
Thank you. Your observation is very interesting. From my experience, and from reading the few studies that exist, I believe it cannot be applied. Therefore, we highlight this factor as a limitation of our study and an interesting approach to be considered in future protocols.
The results are partially expectable since the effect of BAT activity on energy expenditure is well established.
We appreciate your comment and partially agree with the reviewer. In fact, the effect on energy expenditure is in line with previous studies. However, the BAT activation studies showed greater lipid catabolism and not protein catabolism, which was an unprecedented result.
Reviewer 3 Report
Comments and Suggestions for Authors
The study evaluated the effects of caffeine supplementation on energy expenditure and substrate oxidation before and after high intensity interval training in males with high versus low brown adipose tissue activity. This is an interesting study, but I have a number of suggestions for revision:
Line 27: The title of the manuscript mentions caffeine but the aims stated in the abstract don’t mention anything about caffeine.
Line 29: Given the lack of research on females in the are of exercise physiology, restriction of the study to only male participants is a limitation.
Line 31: It is unclear what you mean by “quasi-experimental” here
Lines 33-34: You need to make sure to define all abbreviations here
Line 39: “carbohydrates (CHO; g/day), lipids (FAT; g/day) and proteins (PTN; g/day) were measured”
- Did you mean carbohydrate, lipids, and proteins oxidation here?
- You mention these were measured as g/day, but later in the abstract you present fat oxidation as g/min.
Lines 43-45: “The HBAT-CAF condition, minutes 40, 50, and 60 showed a significant difference for PTN catabolism compared to the other measurement times” What was this difference? Was protein catabolism higher or lower in the HBAT-CAF condition?
Line 45: The conclusion statement in your abstract does not accurately reflect the results presented in the abstract.
I suggest adding the dose of caffeine use in the abstract.
Line 54: In this opening sentence you mention how overweight and obesity are problems, but your participants were described as “highly trained”. This opening sentence therefore does not reflect the topic of your manuscript.
Lines 74 and 75: Validity and reliability are different concepts. Here you have implied they are the same thing.
Throughout the introduction you should define abbreviations the first time you use them.
Line 102: Again, clarify what you mean by “quasi-experimental”.
Was the study registered with a clinical trials registry?
Please make sure the study complies with CONSORT guidelines for cross-over studies.
Line 105: This is the first time you mention randomization. I suggest mentioning this in the study design and in the abstract.
Lines 137-138: This needs to be translated to English. What was the approximate mg of caffeine consumed by participants per day?
Line 176: Why did you use a different temperature and relative humidity for this portion of the experiment compared to that described on line 148?
Line 219: “PTN catabolism was calculated via the analyzer’s proprietary software” – Has this been validated against other techniques?
Figure 4 legend: Please indicate what the error bars represent (i.e., SD, SE?).
I don’t think Table 1 is really needed, as Figure 4 presents the important results for this variable.
Lines 286-313: This section presents differences in protein oxidation in detail. I think this section can be shortened by highlighting the most important results and perhaps referring the reader to the table for other results.
Author Response
The study evaluated the effects of caffeine supplementation on energy expenditure and substrate oxidation before and after high intensity interval training in males with high versus low brown adipose tissue activity. This is an interesting study, but I have a number of suggestions for revision:
We sincerely appreciate the time and effort you put into reviewing our manuscript. Your feedback helped us identify key areas for improvement.
Line 27: The title of the manuscript mentions caffeine but the aims stated in the abstract don’t mention anything about caffeine.
Thank you. We accept 1st reviewer’s suggestion to change the title. We also modified the Abstract objective to include caffeine, and new results based the suggestion of the 1st review.
Thermogenic activation of adipose tissue by caffeine during strenuous exercising and recovery: a double-blind crossover study
Line 29: Given the lack of research on females in the are of exercise physiology, restriction of the study to only male participants is a limitation.
Thank you. We insert this limitation.
While this study focused on male physically active individuals, future research should prioritize populations with obesity or overweight to better align findings with weight loss applications. It will also be interesting if future protocols measure people of different ages and women as well.
Line 31: It is unclear what you mean by “quasi-experimental” here
We removed it from the Abstract due to the limited number of words in this section and kept it in the method, where we explain what makes this protocol a quasi-experimental design.
Lines 33-34: You need to make sure to define all abbreviations here
Thank you very much for the observation, all the acronyms in the Abstract have been explained.
Line 39: “carbohydrates (CHO; g/day), lipids (FAT; g/day) and proteins (PTN; g/day) were measured”
Did you mean carbohydrate, lipids, and proteins oxidation here?
You mention these were measured as g/day, but later in the abstract you present fat oxidation as g/min.
Thank you for the observation; it was indeed a flaw noted by all reviewers. Our study measures in g/min, as do the previous studies we cited; we corrected this throughout the text. Furthermore, for clarity in the Abstract, we indicated that the macronutrient catabolism rate was measured by spirometry.
Lines 43-45: “The HBAT-CAF condition, minutes 40, 50, and 60 showed a significant difference for PTN catabolism compared to the other measurement times” What was this difference? Was protein catabolism higher or lower in the HBAT-CAF condition?
Thank you for your observation. Because we modified the analysis methods according to reviewer 1's recommendations, all results were modified, which in turn required us to rewrite the Abstract.
Line 45: The conclusion statement in your abstract does not accurately reflect the results presented in the abstract.
We agree with the reviewer. As with the previous correction, the entire Abstract has been rewritten, as have the study's conclusions.
I suggest adding the dose of caffeine use in the abstract.
We agree with the reviewer. This information was added in absolute and relative values to body mass.
Line 54: In this opening sentence you mention how overweight and obesity are problems, but your participants were described as “highly trained”. This opening sentence therefore does not reflect the topic of your manuscript.
We agree with the reviewer. In fact, it was very discrepant; this was an observation made by all reviewers of the manuscript. We removed all information regarding obesity and overweight and suggested at the end of the discussion that this be a focus for future studies.
While this study focused on male physically active individuals, future research should prioritize populations with obesity or overweight to better align findings with weight loss applications. It will also be interesting if future protocols measure people of different ages and women as well.
Lines 74 and 75: Validity and reliability are different concepts. Here you have implied they are the same thing.
We agree with the reviewer and rewrote part of the text where we talk about the feasibility of IRT to estimate BAT activity.
IRT has appeared as a promising non-invasive method for assessing BAT activity through the measurement of supraclavicular skin temperature (SST) [10-12]. IRT quantifies heat emission (a proxy for BAT thermogenesis) before, during, or after cold or pharmacological stimulation, with agreement against PET/CT confirming its reliability [10,11]. A systematic review indicates that SST is increasingly recognized as a surrogate marker for BAT volume and function, with 18 out of 24 studies linking SST changes to human BAT activity [12]. This approach leverages BAT’s inherent capacity for heat generation via non-shivering thermogenesis, a process detectable as localized surface warming [6,12,13]. Specifically, SST over the supraclavicular region correlates positively with BAT activity, especially during cold exposure where heightened metabolic activity elevates local skin temperature [12,14,15]. Crucially, IRT offers a comprehensive, cost-effective alternative to traditional techniques like 18F-FDG PET, which estimates BAT activation via glucose uptake [14]. Its non-invasive nature further enables repeated assessments, providing dynamic insights into BAT behavior across physiological and pathological states [13]. Consequently, IRT stands as a valuable tool for evaluating BAT’s role in metabolism, allowing clinicians and researchers to monitor changes without invasive procedures [14,15].
Throughout the introduction you should define abbreviations the first time you use them.
Thank you for your observation, we have revised the introduction and explained all the acronyms.
Line 102: Again, clarify what you mean by “quasi-experimental”.
Thank you for your observation, we have inserted this explanation at the beginning of the methods.
This study employs a quasi-experimental design because participants were pre-classified into HBAT and LBAT groups based on their basal brown adipose tissue activity, which is an intrinsic variable not subject to randomization. Although the intervention (CAF/PLA) was randomized and controlled, the primary exposure (BAT status) was determined observationally. This protocol was previously approved by ethical committee of Federal University of Juiz de Fora (CAAE: 35605220.1.0000.5147, register number: 4.366.750) and registered in the Brazilian clinical trials network – REBEC.
Was the study registered with a clinical trials registry?
Thank you, your observation is important. The project has been registered with the Brazilian Clinical Trials Network, but we have not yet received the registration number. However, I believe it will be published by the publication date of this manuscript, if accepted. We have attached the new version of the manuscript to the registration receipt and have made a reference to it in the text.
Please make sure the study complies with CONSORT guidelines for cross-over studies.
We thank the reviewer, we adopted the CONSORT guidelines and added the checklist as an attachment.
Line 105: This is the first time you mention randomization. I suggest mentioning this in the study design and in the abstract.
Thank you for this observation. Done.
Methods: In this randomized double-blind crossover study, 35 highly-trained males (HBAT-CAF, HBAT-PLA, LBAT-CAF, LBAT-PLA) performed 30-min treadmill HIIT. BAT activity was assessed via infrared thermography. Breath-by-breath ergospirometry measured EE (kcal/min), carbohydrates (CHO), lipids (LIP), and protein (PTN) oxidation. Temporal patterns were analyzed using 2nd/3rd-order polynomial regression.
Lines 137-138: This needs to be translated to English. What was the approximate mg of caffeine consumed by participants per day?
Thank you for this observation. We reviewed as suggested.
Line 176: Why did you use a different temperature and relative humidity for this portion of the experiment compared to that described on line 148?
Thank you for highlighting this important detail regarding experimental conditions. The difference in temperature (and relative humidity) between 1st day (19°C) described on line 148 and the subsequent phases (21°C) on line 176 reflects distinct protocol requirements. 19°C Phase (Line 148): This lower temperature was intentionally employed as a standardized cold stimulus specifically to activate Brown Adipose Tissue (BAT). 21°C (Line 176): The slightly higher temperature of 21°C was used for the exercise portion of the protocol. This temperature represents a standard thermoneutral condition for resting humans and is optimal for conducting exercise interventions
Line 219: “PTN catabolism was calculated via the analyzer’s proprietary software” – Has this been validated against other techniques?
We appreciate the reviewer's important methodological question regarding the validation of protein (PTN) catabolism calculations. While Frayn’s stoichiometric equations are widely accepted for carbohydrate and lipid oxidation, protein catabolism estimation presents unique challenges. The Metasoft® Studio software estimates protein catabolism based on urinary urea nitrogen excretion (when available) integrated with respiratory exchange ratio (RER) data. This aligns with the equation: Protein oxidation (g/min) = 6.25 × urinary N excretion (g/min). To improve the description, we have modified this paragraph
Following the HIIE session, participants rested in a supine position for 30 minutes while recovery metabolism was assessed via indirect calorimetry. Oxygen consumption and carbon dioxide production were continuously monitored throughout the experiment to estimate total energy expenditure (EE; Kcal) and substrate utilization rates (g/min) for carbohydrate (CHO), lipids (FAT) and proteins (PTN). CHO and FAT oxidation rates were derived using Frayn’s stoichiometric equations as in Alcantara, et al. [31]. Protein (PTN) catabolism was estimated using the manufacturer’s protocol (Metasoft® Studio 5.5.1, Cortex, Leipzig, Germany), which implements Weir’s stoichiometric equations based on urinary urea nitrogen excretion [32]. This approach is consistent with established methodologies for indirect protein oxidation assessment [33].
Figure 4 legend: Please indicate what the error bars represent (i.e., SD, SE?).
All Figures have been modified to meet 1st reviewer recommendations.
I don’t think Table 1 is really needed, as Figure 4 presents the important results for this variable.
The Table was removed for the same reason answered above, we inserted a new Table with the main findings of the model.
Lines 286-313: This section presents differences in protein oxidation in detail. I think this section can be shortened by highlighting the most important results and perhaps referring the reader to the table for other results.
All results were modified due to the change in data analysis. In fact, they became more concise.
Reviewer 4 Report
Comments and Suggestions for Authors
Dany Alexis Sobarzo Soto and colleagues reported the effect of caffeine vs placebo supplementation on metabolism in high and low brown adipose tissue athletes during and after one session of high intensity interval exercise.
The same group has previously reported the effect of caffeine on increasing energy expenditure in high vs low BAT individuals.
In this study they have added an exercise component to the study. Although the study design is good and proper controls were taken, the effect of an acute bout of exercise on CAF+BAT-induced changes in metabolism is a stretch, especially considering the scope of obesity.
Also, the study did not report any blood or plasma markers like catecholamines, FFAs, etc.
The study only included healthy, trained individuals, which lowers the generalizability to sedentary, especially obese individuals.
The tables should be converted to figures for better visibility and understanding. Tables should be kept in the supplementary files.
Specific-
Lines 137-138 - The sentence should be in English. Please correct.
Author Response
Reviewer 4
Dany Alexis Sobarzo Soto and colleagues reported the effect of caffeine vs placebo supplementation on metabolism in high and low brown adipose tissue athletes during and after one session of high intensity interval exercise.
The same group has previously reported the effect of caffeine on increasing energy expenditure in high vs low BAT individuals.
We thank you for your thorough and constructive review of our manuscript. Your insights have been invaluable in identifying critical improvements. This is our second study along these lines, we will soon have a chronic study, which is currently underway.
In this study they have added an exercise component to the study. Although the study design is good and proper controls were taken, the effect of an acute bout of exercise on CAF+BAT-induced changes in metabolism is a stretch, especially considering the scope of obesity. Also, the study did not report any blood or plasma markers like catecholamines, FFAs, etc. The study only included healthy, trained individuals, which lowers the generalizability to sedentary, especially obese individuals.
As noted by the other reviewers, our study was inconsistent because we discussed obesity, yet we evaluated athletes. We reviewed everything and removed anything related to obesity. We recommend it as a possible future protocol, which could be based on the data presented here. We also modified the discussion to be more aligned with our results and not with measures we did not take.
The tables should be converted to figures for better visibility and understanding. Tables should be kept in the supplementary files.
The first reviewer suggested modifying the data analysis method, so all Tables and Figures were modified.
Specific-Lines 137-138 - The sentence should be in English. Please correct.
Thank you for this observation. We translate this statement.
Round 2
Reviewer 1 Report
Comments and Suggestions for Authors
Thanks for having accepted my suggestions, the manuscript improved substantially
Reviewer 4 Report
Comments and Suggestions for Authors
Authors have satisfactorily revised the manuscript as per the suggestions. Manuscript can be accepted as such.